# Cumulus cell expansion, nuclear maturation and embryonic development of bovine cumulus-oocyte complexes matured in varying concentrations of follicular fluid

**Verónica M. Negrón-Pérez**[1¤], **Abdullah Al Naib**[1], **Abigail L. Zezeski**[1], **Victoria L. McCracken-Harlow**[1], **George A. Perry**[2], **Alan D. Ealy**[1], **Michelle L. Rhoads**[1]*

**1** School of Animal Sciences, Virginia Tech, Blacksburg, Virginia, United States of America, **2** Texas A&M AgriLife Research and Extension Center, Overton, Texas, United States of America

¤ Current address: Department of Animal Sciences, University of Puerto Rico, Mayagüez, Puerto Rico
* rhoadsm@vt.edu

## Abstract

In this study, we tested the overall hypothesis that CC expansion and early embryo development would be improved by including follicular fluid (FF) from small or large follicles in the oocyte maturation medium. In the first experiment, FF aspirated from bovine abattoir ovaries was added to the maturation medium at 0, 25, 50, 75 or 100%. Images of individual COCs were captured at 0, 6, 12 and 19 hours (h) of the maturation period and analyzed to calculate change in the total area over time. Cumulus cell expansion was greatest in COCs matured in 75% and 50% FF, and these differences were detectable at 12 (75% FF only) and 19 h (50% and 75% FF) of maturation. The improvement in CC expansion was greatest when FF from small follicles was used. Treatments for the subsequent experiments were selected based upon the results of the first experiment. Oocyte nuclear maturation rates were observed after supplementing the maturation medium with 0 or 75% FF and maturing for 19 h. The rate of nuclear maturation as determined by the presence or absence of the first polar body was similar between control (0% FF) and treated (75% FF) groups. In the final experiment, COCs were matured in 0%, 50% or 75% FF in preparation for IVF. Duration of the maturation period (12, 19 or 22 h) and size of the follicles from which FF was collected (small or large) also varied. In general, FF supplementation at 50% did not affect the zygotes' developmental potential (neither increased nor decreased). Supplementation of maturation medium with 75% FF from small follicles consistently reduced measures of embryo development while 75% FF from large follicles yielded mixed results. It is concluded that FF supplementation improves CC expansion, but the greater CC expansion does not benefit subsequent embryo development. Notably, however, the 50% FF treatment did not reduce blastocyst rates, indicating that FF can be included in maturation media at concentrations of 50% or less with no detriment to IVF outcomes.

**Data Availability Statement:** Data underlying this manuscript are made accessible through the

Virginia Tech Data Repository at https://doi.org/10.7294/27157005.

**Funding:** Funding for this work was provided, in part, by the Virginia Agricultural Experiment Station and the Hatch Program of the National Institute of Food and Agriculture, U.S. Department of Agriculture. The funders had no role in study design, data collection and analysis, decision to publish, or preparation of the manuscript.

**Competing interests:** The authors have declared that no competing interests exist.

## Introduction

Early embryonic death in utero and associated low pregnancy rates are sources of significant financial loss in the cattle industry [1, 2]. These losses are greater for pregnancies resulting from *in vitro* produced embryos [3–5] and one of the reasons for their failure is the low quality of the fertilized oocyte from which they are derived [6, 7]. For *in vitro* maturation and fertilization of bovine oocytes, cumulus oocyte complexes (COCs) are removed from small- to medium-sized tertiary follicles and then selected based upon visual appraisal of their quality. They are then placed into a maturation medium where they undergo spontaneous, luteinizing hormone (LH)-independent meiotic maturation [8] and cumulus cell (CC) expansion in preparation for fertilization. They must successfully complete the process of maturation, which is necessary for subsequent fertilization. Nuclear maturation results in the completion of meiosis I, which causes a reduction in chromosome number through extrusion of the first polar body. During maturation, the CCs surrounding the oocyte expand, and previous research has shown that this expansion is important for successful fertilization [9, 10].

*In vivo*, bovine oocyte maturation and CC expansion occur in the dominant follicle after the surge of LH but prior to ovulation. At this time, the bovine COC is maturing in an enclosed follicle where it is bathed in follicular fluid (FF). Unfortunately, however, the importance of FF for maturation and CC expansion is not yet understood. Research has shown that, in comparison to *in vivo*-matured counterparts, *in vitro*-matured COCs are less likely to develop to the blastocyst stage [11]. The absence of FF in the *in vitro* maturation system may be at least partially responsible for the apparent reduction in oocyte competence.

The FF surrounding the oocyte is composed of a variety of proteins [12], hormones, extracellular vesicles [13], and other components that make up the complex microenvironment in which the oocyte develops. The usefulness of the inclusion of FF in maturation media has been researched and debated for decades. Theoretically, it makes sense for FF to be a component of maturation media because that is the fluid in which COC maturation would occur *in vivo*. Unfortunately, however, there are many caveats associated with the incorporation of FF into maturation media. Pooled FF is undefined and potentially extremely variable. These attributes have likely contributed to the variation in the results of previously published studies where COCs were matured in media supplemented with FF. Despite these limitations, there are circumstances under which analyses of FF and its use in oocyte maturation media are advantageous. Characteristics of FF have been directly linked with the developmental competence of their associated oocytes and maturation of "neutral" oocytes in FF from animals and humans under experimental or clinical protocols has been used as an informative assay [14–17]. Beyond these practical applications, inclusion of FF in maturation media is a direct approach for studying the intricate communication between the COC and components of its preovulatory microenvironment.

In previous work, follicular fluid has been added to maturation media at concentrations ranging from as little as 1% [18] to as much as 100% [19] and can have positive or negative effects on oocyte maturation and blastocyst formation based on factors such as the age of the animal from which the fluid was collected [20]. In these previous studies, rates of blastocyst development (and sometimes cleavage rates) were used as the benchmarks to determine the success or failure of the respective treatments. Unfortunately, however, this approach does not provide a clear indication of how or when FF may be helping or hindering the *in vitro* maturation process. A systematic analysis of additional indicators of maturation at different concentrations of FF will improve the current understanding of the role and efficacy of FF in the COC maturation process.

Therefore, the objectives of this study were three-fold. We first sought to assess the CC response to inclusion of FF in maturation medium over the course of the maturation period

(measured as degree of expansion). We then tested whether the concentration of FF optimal for CC expansion also improved oocyte nuclear maturation and/or early embryo development. It was hypothesized that inclusion of FF in the oocyte maturation medium would increase CC expansion area and improve oocyte nuclear maturation and the ability of the embryo to reach the blastocyst stage. These objectives were addressed with a series of experiments in which the area of the COCs was systematically measured, nuclear maturation was observed and the matured oocytes were fertilized *in vitro* for assessment of blastocyst development.

## Materials and methods

### Collection and preparation of follicular fluid

Animal tissues were used in this work, but because of their source, no ethical approval was required as per the guideline set forth by the Virginia Tech Institutional Animal Care and Use Committee. Bovine ovaries were purchased from a local slaughterhouse (Brown Packing, Gaffney, SC) in the months of September-May to avoid any potential detrimental effects of summer heat stress on COC quality. The ovaries were washed and transported in 0.9% saline solution supplemented with penicillin (100 units/ml) and streptomycin (100 μg/ml). Upon arrival, FF was aspirated from either small (3–5 mm diameter) or large (8–10 mm diameter) antral follicles. The fluid was pooled or kept separate, filtered through a 0.2 μm cell strainer and stored at -20°C for future use.

### Maturation medium preparation

Unless otherwise stated, all chemicals were obtained from Sigma-Aldrich (St. Louis, MO, USA). A standard oocyte maturation medium (OMM) formula was used for control oocyte maturation medium [cOMM; Tissue Culture Medium-199 with Earle's salts supplemented with 10% FBS, 50 μg/ml gentamicin (Gibco, Grand Island, NY, USA), 1 mM glutamine, 40 μg/ml FSH (Folltropin-V; Bioniche, Belleville, ON, Canada), 1 mg/ml E2 (Sigma-Aldrich) and 10 μg/ml EGF (Sigma-Aldrich)]. A separate medium was used as the base for all FF treatments and was designated experimental oocyte maturation medium (eOMM). The eOMM consisted of Tissue Culture Medium-199 with Earle's salts supplemented with 10% (v/v) charcoal stripped FBS, 50 μg/ml gentamicin and 1 mM glutamine. Estradiol, FSH and EGF were absent from this base medium. Thus, the standard oocyte maturation medium was only used in the control groups (cOMM) and the eOMM was used as base for all the treatment groups.

### Oocyte collection and maturation

Cumulus-oocyte-complexes were obtained by bisecting 3–8 mm follicles using a scalpel and by washing in oocyte collection medium [Tissue Culture Medium-199 with Hank's salts supplemented with 2% FBS, penicillin (100 units/ml) and streptomycin (100 μg/ml), 2 mM glutamine and heparin ($\geq$0.018 USP units/ml)]. Only those with homogeneous cytoplasm and more than three layers of compact CCs were selected for experimental use. Specific conditions for maturation were dependent on each experiment and are described below. Abbreviations for all treatments are defined in S1 Table.

### Experiment 1: Cumulus cell expansion following maturation with varying concentrations of follicular fluid

Selected COCs (n = 1321) were matured individually on untreated 100 mm culture plates (ThermoFisher, Waltham, MA, USA), in 10 μl drops of control or supplemented eOMM overlaid with mineral oil (EmbryoMax; Millipore, Billerica, MA, USA), in a humidified gas

atmosphere of 5% (v/v) $CO_2$ and 19% (v/v) $O_2$ at 38.5˚C. The treatment groups included eOMM supplemented with 0, 25, 50, 75 and 100% pooled FF (i.e. mixture of small and large FF). The total number of COCs per treatment (0, 25, 50, 75 and 100% FF) was 129–176 over 16 replicates (n = 14–30 COCs/treatment/replicate), and COCs were matured for 19 h. Digital images of each COC were captured at 0, 6, 12 and 19 h using an inverted microscope (EVOS XL; AMG, Mill Creek, WA, USA). The area of each COC was quantified from the images using ImageJ software v1.47 (NIH, Bethesda, MD, USA). Percent of expansion increase [(area at 19 h–area at 0 h)/area at 0 h *100] was calculated for each COC and only COCs that expanded greater than 10% were included in the analysis (n = 782).

In order to differentiate the effects of FF from small compared to large follicles, another subset of COCs (n = 300) were individually matured in 10 μl drops of control or supplemented eOMM. Treatment groups included eOMM supplemented with 0, 50 and 75% FF collected from either small or large follicles. The COCs were matured for 22 h under the atmospheric conditions described above. Digital images were captured at 0 and 22 h. The area of each COC was quantified from the images using ImageJ software. A total of 60 COCs per treatment (0, 50 and 75% of either small or large FF) were observed over 6 replicates.

## Experiment 2: Oocyte nuclear maturation with or without follicular fluid

In order to determine the effects of pooled FF on oocyte nuclear maturation, COCs (n = 198; 32–160 COCs/replicate) were individually matured as described above for 19 h in either 0% or 75% pooled FF-supplemented eOMM. Oocytes were then denuded by vortexing in hyaluronidase (1000 U/ml in ~0.5 ml HEPES-TALP), washed in Phosphate-buffered saline (PBS) containing 0.1% (w/w) polyvinylpyrrolidone (PVP) and fixed in 10% neutral buffered formalin (ThermoFisher, Waltham, MA) for 15 minutes at room temperature. After fixation, they were washed three times in PBS supplemented with 0.1% Tween 20 (ThermoFisher) and 1 mg/ml BSA (Fraction V). Permeabilization was conducted by exposing the oocytes to PBS supplemented with 0.25% Triton X-100 (ThermoFisher) for 1 h. Oocytes underwent DAPI staining (NucBlue, Invitrogen, Carlsbad, CA, USA) for 15 minutes to determine whether or not the first polar body had been extruded from each individual oocyte. Images were captured with a Nikon Eclipse Ti fluorescence microscope (Melville, NY, USA).

## Experiment 3: In vitro production of embryos following oocyte maturation with or without follicular fluid

All bovine IVF procedures were based on previously described protocols [21–25]. Selected COCs were placed in groups of 10 COCs per 50 μl drop of standard cOMM or eOMM supplemented with 75% pooled FF (i.e. mixture of small and large FF), covered with mineral oil and placed in a humidified gas atmosphere of 5% (v/v) $CO_2$ and 19% (v/v) $O_2$ at 38.5˚C. The duration of the maturation period was 12, 19 or 22 h. Upon completion of maturation, COCs from the same treatment were pooled, washed in HEPES- Tyrode's albumin lactate pyruvate [HEPES-TALP; HEPES-TL (Caisson Laboratories, Inc; North Logan, UT, USA) supplemented with 3 mg/ml BSA (Fraction V), 22 μg/ml sodium pyruvate and 75 μg/ml gentamicin] and fertilized in plates containing 1.7 ml of IVF-TALP [IVF-TL (Caisson Laboratories) supplemented with 6 mg/ml BSA (essentially fatty acid free), 22 μg/ml sodium pyruvate, 10 μg/ml heparin and 50 μg/ml gentamicin]. Two frozen-thawed semen straws from different *B. taurus* bulls were pooled, purified with BoviPure-BoviDilute 40% [v/v and 80% (v/v)], and diluted to a final concentration in the fertilization dishes of 1 x $10^6$/ml. Fertilization time was 18–22 h in a humidified gas atmosphere of 5% (v/v) $CO_2$ and 19% (v/v) $O_2$ at 38.5˚C for all groups. Presumptive zygotes were collected, exposed to hyaluronidase (1000 U/ml in ~0.5 ml

HEPES-TALP) and vortexed for 5 min to remove the CCs. Then, presumptive zygotes were washed three times in HEPES-TALP and placed in groups of 25–30 zygotes per 50 μl drop of synthetic oviductal fluid–bovine embryo 2 (SOF-BE2) [26] covered with mineral oil in a humidified gas atmosphere of 5% (v/v) $CO_2$, 5% (v/v) $O_2$ and the balance nitrogen, at 38.5˚C. Cleavage rates were assessed on day 3 and blastocyst rates on day 8 post-fertilization. This experiment was repeated 8 times with 220–557 zygotes for each treatment (total n = 2159).

In four subsequent replicates, selected COCs were matured in eOMM containing 0%, 50% or 75% FF collected from small or large follicles (115–157 COCs per group; total n = 692). The duration of the maturation period was 22 h. Upon maturation, COCs were fertilized as described above and cultured until day 8 post-fertilization.

## Statistical analysis

The SAS v 9.4 software package (SAS Institute Inc., Cary, NC, USA) was used for statistical analysis. Data were analyzed for main effects of FF treatment, maturation time (where appropriate), replicate study and their interactions using the mixed procedure of SAS. For experiment 1, the dependent variable was the size of the COC (*i.e.* CC expansion) for each time point when images were captured and for experiment 2 the dependent variable was presence or absence of a polar body. For experiment 3, day 3 cleavage rate was calculated as the number of zygotes that cleaved divided by the total number of presumptive zygotes and, day 8 blastocyst rate was calculated as the number of zygotes that became blastocysts divided by the total number of presumptive zygotes. At day 8, the number of zygotes that became blastocysts divided by the total number of zygotes that cleaved was also calculated and analyzed. Separation of means was conducted with the Tukey procedure of SAS. Results are reported as least squares means ± standard error of the means (SEM). Statistical significance was declared at P<0.05.

## Results

### Experiment 1: Cumulus cell expansion following maturation with varying concentrations of follicular fluid

Follicular fluid supplementation increased CC expansion over the 19 h maturation period (Table 1 and Fig 1). At the beginning of the experiment (0 h), there were no differences in COC size between treatments (Table 1). Within treatments, the first detectable increase in COCs area was observed at 6 h of maturation but only in those COCs matured in 50% and 75% FF (P<0.01). The COCs matured in 0%, 25% and 100% FF first increased (P<0.01) in size at the 12 h measurement. When comparing across treatments, COCs matured in the 75% FF were greater in size (P<0.05) than those in all other treatments at the 12 h and 19 h

**Table 1. Area of cumulus oocyte complexes (COCs; μm²) at 0, 6, 12 and 19 h of the maturation period.** Cumulus oocyte complexes were matured in media containing 0%, 25%, 50%, 75% or 100% pooled follicular fluid. Data presented as least squares mean ± SEM (x100000).

| Treatment | Total COCs | 0 h | 6 h | 12 h | 19 h |
|---|---|---|---|---|---|
| 0% (cOMM) | 166 | 1.25 ± 0.05[a] | 1.44 ± 0.05[a] | 2.12 ± 0.07[bxy] | 3.01 ± 0.11[cx] |
| 25% | 149 | 1.23 ± 0.05[a] | 1.44 ± 0.08[a] | 2.02 ± 0.12[bx] | 2.83 ± 0.18[cx] |
| 50% | 162 | 1.28 ± 0.05[a] | 1.64 ± 0.07[b] | 2.57 ± 0.11[cy] | 3.78 ± 0.18[dy] |
| 75% | 176 | 1.36 ± 0.05[a] | 1.87 ± 0.08[b] | 3.11 ± 0.13[cz] | 4.75 ± 0.19[dz] |
| 100% | 129 | 1.04 ±0.05[a] | 1.36 ± 0.09[a] | 2.06 ± 0.16[bxy] | 2.71 ± 0.15[cx] |

[abcd]Values within row with different superscripts are significantly different (P<0.01).

[xyz]Values within column with different superscripts are significantly different (P<0.05).

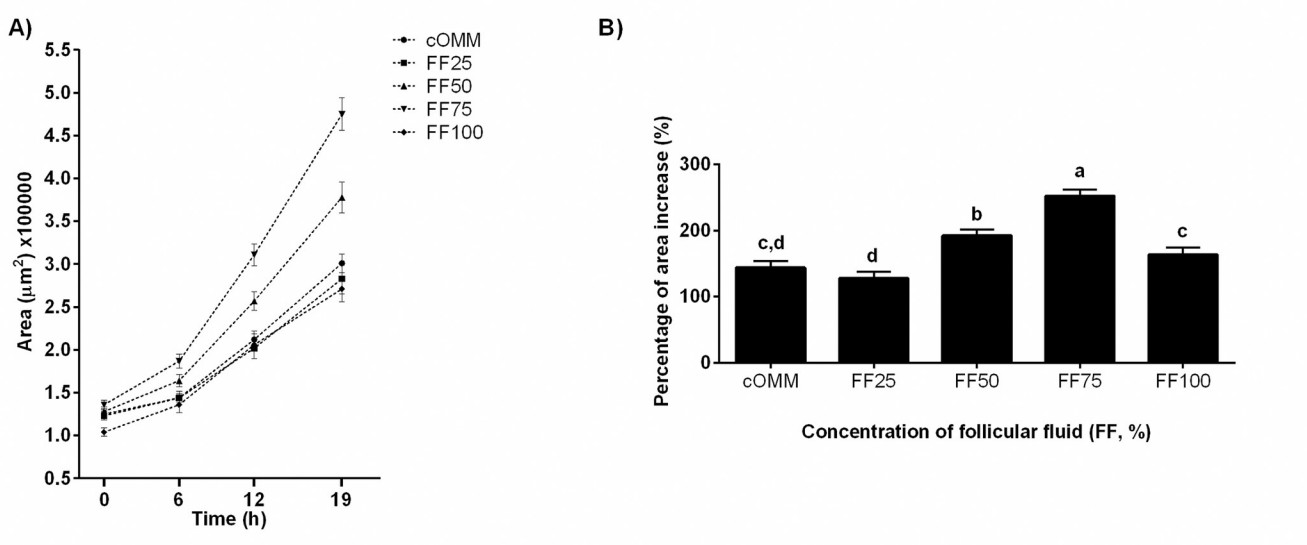

**Fig 1. Effect of follicular fluid (FF) supplementation during the maturation period on cumulus cell expansion.** (A) Mean area of cumulus oocyte complexes (COCs; n = 782; μm$^2$) at 0, 6, 12 and 19 h of the maturation period. Cumulus oocyte complexes were matured in oocyte maturation medium supplemented with 0% (cOMM), 25% (FF25), 50% (FF50), 75% (FF75) or 100% (FF100) of pooled FF. (B) Increase (%) in the area of COCs (n = 782) after maturation in designated concentrations of FF for 19 h. Data are presented as least squares mean ± SEM of the percentage of area increase from 0 h to 19 h. Different letters above the bars indicate significant differences between treatments (P≤0.03).

observations (Fig 1A and Table 1). By the end of maturation (19 h), COCs exposed to 50% and 75% pooled FF had greater percent increase (P≤0.03 and P<0.01, respectively; Fig 1B) compared to other treatments with the largest increase corresponding to the 75% treated group. When the experiment was repeated with 50% and 75% FF from large or small follicles, all FF treatments improved CC expansion over the control medium (regardless of the FF concentration and size). This effect was greatest when 50% and 75% FF from small follicles was used (Fig 2).

## Experiment 2: Oocyte nuclear maturation with or without follicular fluid

The rate of nuclear maturation was similar between treatments regardless of maturation time (19 h or 22 h) or FF supplementation. When COCs were matured in 0% FF, 62% exhibited an extruded polar body. When matured in the 75% FF treatment, 56% of the COCs contained an extruded polar body.

## Experiment 3: In vitro production of embryos following oocyte maturation with or without follicular fluid

Embryo cleavage and blastocyst rates were not improved by the inclusion of FF in the maturation media (Figs 3 and 4). There were also no interactions of treatment (0% vs 75% FF) and maturation time (Fig 3). The shortest maturation interval severely reduced cleavage and blastocyst rates regardless of inclusion of FF in the maturation medium (Table 2). Blastocyst development calculated as the percentage of zygotes becoming blastocysts at day 8 (Fig 3B) and as the percentage of cleaved embryos becoming blastocyst (Fig 3C) was decreased when the embryos originated from COCs exposed to 75% pooled FF.

Compared to the control group, the percentage of putative zygotes that cleaved was lower when the oocytes were exposed to small FF (at both 50% and 75% supplementation) but not

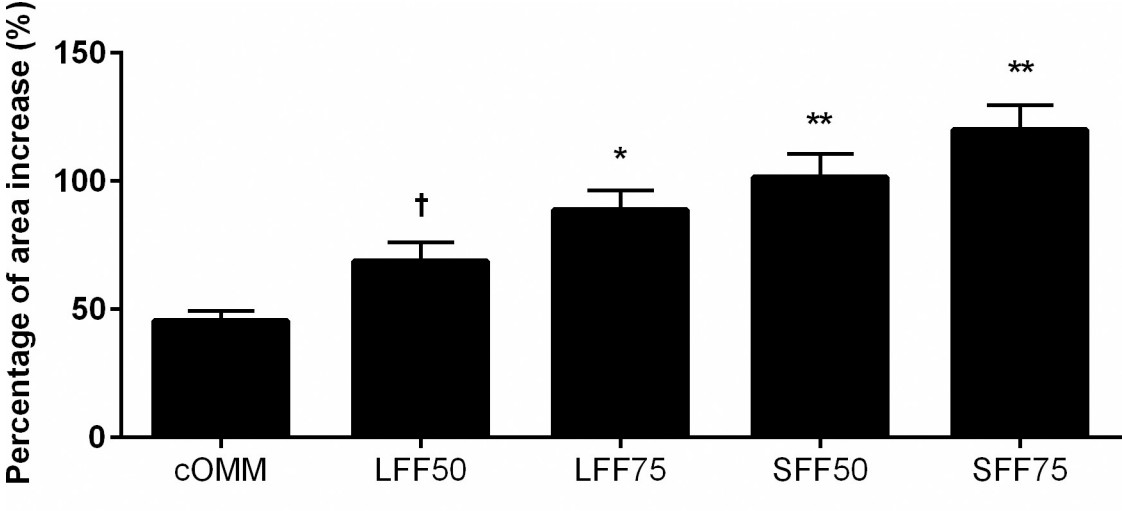

**Fig 2. Percentage of increase in the area of cumulus oocyte complexes (COCs).** Cumulus oocyte complexes (n = 300) were matured in oocyte maturation medium supplemented with 0% (cOMM), 50% or 75% of either small (SFF) or large (LFF) FF for 22 h. Results shown are least-squares means ± SEM. * P<0.05; ** P<0.01.

affected by exposure to large FF ([Fig 4A]). Blastocyst development calculated as the percentage of zygotes becoming blastocysts at day 8 ([Fig 4B]) was decreased when the embryos originated from COCs matured in 75% FF from either small or large follicles. When blastocyst development was calculated as the percentage of cleaved embryos becoming blastocysts ([Fig 4C]), development was reduced by exposure to 75% FF from small follicles, but not affected by 75% FF from large follicles. None of these measurements of blastocyst development were affected by maturation in 50% FF, regardless of follicle size.

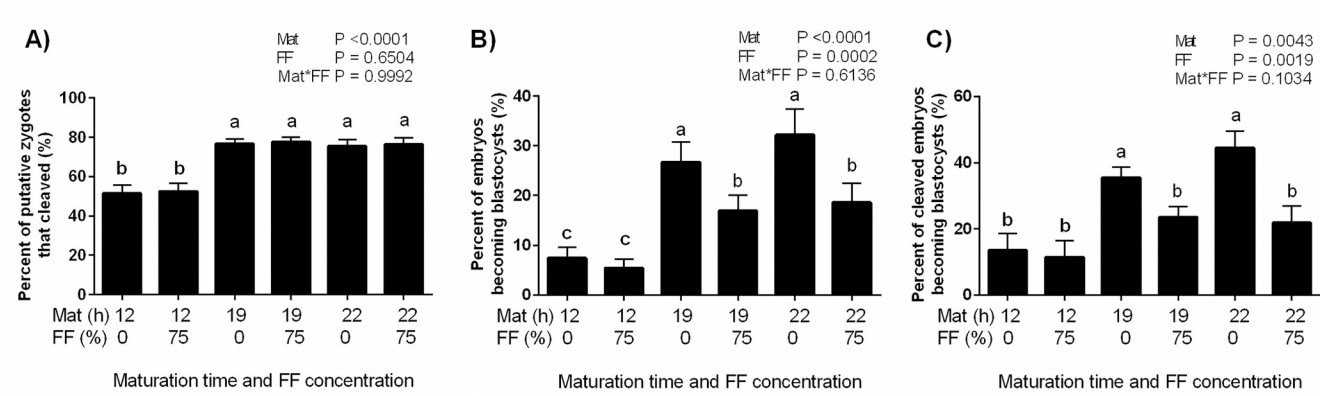

**Fig 3. Effect of follicular fluid (FF) supplementation and time of maturation on development of embryos to the blastocyst stage.** Cumulus oocyte complexes (COCs; n = 2159) were exposed to oocyte maturation medium supplemented with 0% or 75% FF during the 12 h, 19 h or 22 h maturation period. Data is presented as percent putative zygotes that cleaved (A), percent of putative embryos that became blastocysts (B) or percent of cleaved embryos that became blastocysts (C). P values for the main effects of maturation time (Mat, h), FF supplementation (%) and the interaction are shown within graphs. Different letters above the bars indicate significant difference between treatments (P ≤ 0.02).

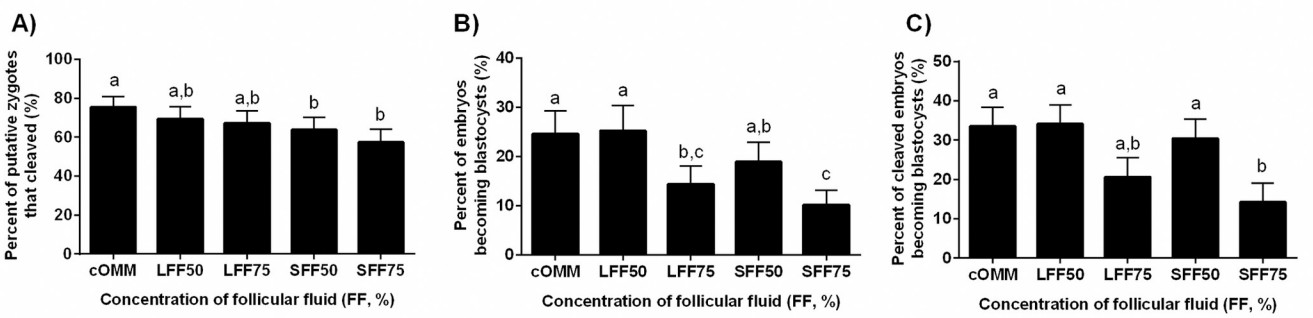

**Fig 4. Effect of supplementation of oocyte maturation medium (OMM) with follicular fluid (FF) from small or large follicles on development of embryos to the blastocyst stage.** Cumulus oocyte complexes (COCs; n = 692) were exposed to OMM supplemented with 0% (cOMM), 50% or 75% FF from small (SFF) or large (LFF) follicles during the 22 h maturation period. Data is presented as percent putative zygotes that cleaved (A), percent of putative embryos that became blastocysts (B) or percent of cleaved embryos that became blastocysts (C). Different letters above the bars indicate significant difference between treatments (P ≤ 0.03).

## Discussion

The efficacy of supplementing maturation media with FF has been examined for several species including humans [27], swine [28, 29], horses [30], sheep [31] and cattle [32–34]. The rate of bovine FF supplementation during maturation has varied from as little as 1% [18] up to 100% [19] but has never been sequentially tested in the manner described here. The current study employed a systematic, quantitative evaluation of the COC before, during and after maturation in media containing sequential concentrations of FF. This systematic approach to evaluating the effects of FF on the COCs revealed that supplementing the OMM with FF increased CC expansion in a manner similar to previous reports [32, 34]. This improvement in expansion was fastest and greatest when the maturation medium contained 75% FF. In fact, by the time the first treatment-specific measurement was collected (6 h of maturation), the COCs in the OMM containing 50% and 75% FF had already begun to expand while the others had not, and by 12 h of maturation the COCs in 75% FF exceeded that of all other treatments. Vigorous expansion continued through 19 h of maturation and did not appear to plateau. By the end of the maturation phase, the size of the COCs in the 75% FF far surpassed the size of the COCs in other treatments. The mechanisms responsible for, and the resulting implications of accelerated CC expansion were not apparent however, and require further study. Nonetheless, this observation is important because the first readily observable indicator of COC maturation is the expansion of the CCs.

Cumulus oocyte complexes that do not undergo expansion are much less likely to undergo cleavage and subsequent development [9]. Although many previous studies have examined the effects of FF on CC expansion, most have relied upon a subjective scale system [35]. Objective

**Table 2. Percent of fertilized cumulus oocyte complexes (COCs) that cleaved and became blastocysts.**

| Treatment* | Total zygotes | Putative zygotes that cleaved (%) | Putative zygotes that become blastocysts (%) | Cleaved embryos that became blastocysts (%) |
|---|---|---|---|---|
| Maturation (h) | | | | |
| 12 | 461 | 52.0 ± 3.7[b] | 6.3 ± 1.6[b] | 12.5 ± 3.9[b] |
| 19 | 1,110 | 77.2 ± 2.1[a] | 21.4 ± 3.4[a] | 29.5 ± 2.3[a] |
| 22 | 588 | 76.0 ± 2.7[a] | 24.8 ± 4.2[a] | 33.2 ± 3.9[a] |

*Main effect of maturation time. Cumulus oocyte complexes were matured for 12, 19 or 22 h.
[ab]Values within column with different superscripts are significantly different (P≤0.01).

measurements of CC expansion similar to the current study are rarely collected [32, 34]. The results of previous work are further limited by the fact that FF is usually supplemented at comparatively low concentrations (5–10%) and CC assessment is generally only conducted at the beginning and end of maturation. Despite these differences in experimental approach, however, the results of previous work [32, 34] agree with those of the current study, finding that inclusion of FF in maturation media consistently improves CC expansion.

The observed improvement in CC expansion suggested that perhaps the COCs matured in the 75% FF were more competent and better prepared for fertilization and development. It also was a cause for concern, however, as accelerated expansion could have also meant that the oocyte was ready for fertilization at an earlier time point during maturation. If this had been the case, the usual 19–22 h maturation period would have been too long, and cleavage and blastocyst rates would have been artificially reduced as a result of aging of the oocyte. To test this, some COCs were subjected to an abbreviated maturation period of 12 h. In the same experiment, maturation durations of 19 h and 22 h were directly compared because previous work had indicated that a 19 h maturation phase is sufficient [36] but slightly less than optimal for bovine COCs [37]. Cleavage and blastocyst rates for 19 h were similar to those for 22 h maturation times. The shortest interval to fertilization (12 h) reduced both cleavage and blastocyst rates, however, regardless of the presence or absence of FF in the maturation media. Thus, it appears that the accelerated expansion of the COCs matured in FF was not related to the time it took for resumption of meiosis and nuclear maturation of the oocyte. In addition to the lack of effect on the timing of nuclear maturation, we also determined there was no improvement in the overall rate of nuclear maturation. Cumulus oocyte complexes were matured in 0% or 75% FF for the full 19 h and then stained for observation of the presence or absence of the first polar body, yielding no differences between treatments.

Similar to nuclear maturation rates, cleavage rates were unaffected by inclusion of FF in the OMM, regardless of concentration (only 50% and 75% FF were tested). In a previous study, when bovine FF was added to maturation medium at a concentration of 5%, cleavage rates improved from 60.4% (control) to 81.0% [34]. Unfortunately, this improvement in development (measured as higher cleavage rate) was not maintained, and by the time the embryos reached the blastocyst stage there was no difference between treatments [34]. In agreement with previously published work [38], the only detriment to cleavage rates in the current study was observed when COCs were matured in FF from small follicles (SFF50 and SFF75). Although we have no specific indication of how or why the SFF reduced cleavage rates, this makes biological sense since *in vivo* COCs would only naturally mature in large follicles.

Interestingly, patterns of blastocyst development were not congruent with the treatment effects on cleavage rates. Rather than differing based upon the size of the follicle from which the FF was collected, blastocyst rates were affected by the overall concentration of FF in the OMM. Cumulus oocyte complexes matured in OMM containing 50% FF yielded blastocysts at rates similar to those matured in the control medium. In contrast, those matured in OMM containing 75% FF were inferior, seldom achieving blastocyst rates of 20%, with the only exceptions being LFF when blastocyst rates were calculated as a percent of cleaved embryos. Blastocyst rates for the control and 75% FF treatments were also similar when maturation time was limited to a 12 h duration, but lack of treatment difference was likely related to the severe reduction in blastocyst rates caused by the abbreviated maturation time. Taken together, these results suggest that while the 75% FF was causing maximal CC expansion, it was detrimental for the developmental competence of the oocyte. Interestingly, although 50% supplementation with FF did not improve CC expansion to the same degree as 75% supplementation, it did not hinder blastocyst development. These findings are congruent with the previous work where blastocyst development was not affected by COC maturation in 10% or 30% FF, while 60% FF

was detrimental [39]. Thus, it is plausible that a lower percentage of FF supplementation might prove to be optimal for blastocyst development, particularly in lower quality oocytes.

While some have demonstrated improvements in blastocyst rates with FF supplementation [18], the results of this experiment agree with previous studies where blastocyst rates were not improved by supplementation of the maturation medium with FF [32, 34]. Variation in results between studies might be explained by any number of differences in the experimental designs, such as the timing of collection of FF used for supplementation of maturation media. When bovine FF was collected 20 h post LH surge, and supplemented at 40% in the maturation medium, blastocyst rates were drastically improved [40]. Although these results are compelling, the level of precision needed to collect FF after the LH surge, but prior to ovulation prohibits its wide-spread implementation. When easily-accessible sources of FF were compared within a single study (slaughterhouse ovaries, cyclic cattle, superovulated cattle), there was generally no difference in IVF outcome [15, 19], indicating that the most readily-available sources of FF yield equivalent outcomes when included in oocyte maturation media. Taken together, most studies agree that even though supplementation with FF does not improve blastocyst rates, under certain conditions and concentrations, it also does not reduce development to the blastocyst stage. Therefore, it appears that the use of FF in maturation media is valid in instances where treatment with FF is desired or advantageous.

Undoubtedly, there is still much to learn about the components of FF and their impacts on the developmental competence of COCs. The composition of FF varies with the stage of the follicle and the physiological state of the animal from which it is collected [41, 42]. Changes in specific hormones and metabolites of the FF have been associated with oocyte or embryo quality [42, 43], highlighting the importance of the microenvironment surrounding the COC. Regarding the inclusion of FF in maturation media, it is likely that optimizing the quality and concentration would be beneficial for blastocyst development. Unfortunately, little is known about which components at which concentrations are capable of reliably improving IVF outcomes. In the current study, we demonstrated that under average conditions (created by pooling FF from slaughterhouse ovaries), COCs matured in up to 50% FF develop to blastocysts at the same rate as controls. This finding could be advantageous in addressing numerous research objectives. For example, FF collected from animals in various physiological states at varying stages of follicle development could be used in an IVF system to determine which profiles and, thus, components are most beneficial for blastocyst development. Likewise, when conducting research experiments, FF collected from animals exposed to different treatments could be used as a biological assay to determine how the treatments are affecting COCs. Ultimately, either of these approaches would provide much-needed information about the characteristics of FF that are important for COC competence.

## Conclusions

The results of this study demonstrate that while 75% FF in maturation media appears to maximize CC expansion, this inclusion rate does not improve nuclear maturation of the COC and is generally detrimental for subsequent embryo development. Instead, the lower supplementation rate of 50% improved CC expansion while also maintaining oocyte competence and early embryo development. Perhaps the most important conclusion that can be taken from these results is even though inclusion of FF in maturation media does not improve embryo development, it can be used in a way that also does not reduce development (i.e., at 50% inclusion rate). This observation agrees with the results of several other studies and establishes the validity of maturation in FF, when necessary or desired to meet experimental objectives, despite its

undefined nature. Taken together however, these results are a stark reminder of our lack of understanding of the intricate interactions between the FF, CCs and oocyte.

## Supporting information

**S1 Table. List of abbreviations and treatments.**
(DOCX)

## Acknowledgments

The authors would like to express their appreciation to Ali Wood for her dedication to this work and contribution to the manuscript.

## Author Contributions

**Conceptualization:** Verónica M. Negrón-Pérez, Abdullah Al Naib, George A. Perry, Alan D. Ealy, Michelle L. Rhoads.

**Data curation:** Verónica M. Negrón-Pérez, Abdullah Al Naib, Michelle L. Rhoads.

**Formal analysis:** Verónica M. Negrón-Pérez, Abdullah Al Naib, Michelle L. Rhoads.

**Funding acquisition:** Michelle L. Rhoads.

**Investigation:** Verónica M. Negrón-Pérez, Abdullah Al Naib, Abigail L. Zezeski, Victoria L. McCracken-Harlow, Michelle L. Rhoads.

**Methodology:** Verónica M. Negrón-Pérez, Abdullah Al Naib, George A. Perry, Alan D. Ealy, Michelle L. Rhoads.

**Project administration:** Verónica M. Negrón-Pérez, Abdullah Al Naib, Michelle L. Rhoads.

**Resources:** Michelle L. Rhoads.

**Software:** Michelle L. Rhoads.

**Supervision:** Verónica M. Negrón-Pérez, Abdullah Al Naib, Michelle L. Rhoads.

**Validation:** Verónica M. Negrón-Pérez, Michelle L. Rhoads.

**Visualization:** Verónica M. Negrón-Pérez, Abdullah Al Naib, Michelle L. Rhoads.

**Writing – original draft:** Verónica M. Negrón-Pérez, Michelle L. Rhoads.

**Writing – review & editing:** Verónica M. Negrón-Pérez, Abdullah Al Naib, Abigail L. Zezeski, Victoria L. McCracken-Harlow, George A. Perry, Alan D. Ealy, Michelle L. Rhoads.

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
