## [Decision Letter · Decision Letter 0]

1 Jan 2025

PONE-D-24-44555Cumulus cell expansion, nuclear maturation and embryonic development of bovine cumulus-oocyte complexes matured in varying concentrations of follicular fluidPLOS ONE

Dear Dr. Rhoads,

Thank you for submitting your manuscript to PLOS ONE. After careful consideration, we feel that it has merit but does not fully meet PLOS ONE’s publication criteria as it currently stands. Therefore, we invite you to submit a revised version of the manuscript that addresses the points raised during the review process.

Kind regards,

Sanaz Alaeejahromi

Academic Editor

PLOS ONE

“Funding for this work was provided, in part, by the Virginia Agricultural Experiment Station and the Hatch Program of the National Institute of Food and Agriculture, U.S. Department of Agriculture.”

“The authors would like to express their appreciation to Ali Wood for her dedication to this work and contribution to the manuscript.  Funding for this work was provided, in part, by the Virginia Agricultural Experiment Station and the Hatch Program of the National Institute of Food and Agriculture, U.S. Department of Agriculture.”

“Funding for this work was provided, in part, by the Virginia Agricultural Experiment Station and the Hatch Program of the National Institute of Food and Agriculture, U.S. Department of Agriculture.”

6. We notice that your supplementary table is included in the manuscript file. Please remove them and upload them with the file type 'Supporting Information'. Please ensure that each Supporting Information file has a legend listed in the manuscript after the references list.

Reviewers' comments:

Reviewer's Responses to Questions

**Comments to the Author**

1. Is the manuscript technically sound, and do the data support the conclusions?

Reviewer #1: Yes

Reviewer #2: Yes

2. Has the statistical analysis been performed appropriately and rigorously? 

Reviewer #1: Yes

Reviewer #2: Yes

3. Have the authors made all data underlying the findings in their manuscript fully available?

Reviewer #1: No

Reviewer #2: Yes

4. Is the manuscript presented in an intelligible fashion and written in standard English?

Reviewer #1: Yes

Reviewer #2: Yes

5. Review Comments to the Author

Reviewer #1: The manuscript is well written. The language is generally clear. However, I think some minor adjustments or corrections need to be made to improve the article.

On line 36 to 38, that is the abstract section, you wrote "In the final two experiments".... Are they actually two separate experiments? I think you need to rewrite that sentence to improve clarity.

I think you need to improve the clarity of line 100 to 105. In line 104, you wrote "and adapted to use as base".... That sentence is not very clear.

Line 224 to 225, That sentence is not necessary in the result section. You have already mentioned the aim in the introduction.

The first two paragraphs of the discussion section are absolutely not necessary. I think you should discuss your findings and briefly explain how they relate or differ from similar studies. You only gave an overview of the literature, and that is something to be done in the introduction section. The same applies to the first few sentences of the third paragraph. The correct discussion only started in line 307. I think you need to remove the first two and half paragraphs in the discussion and improve on the discussion.

Reviewer #2: No comment for the authors. The topic is novel and the languish is sound. The article can be accepted in the current form.

All questions above were answered . No additional comments for the author, including concerns about dual publication, research ethics, or publication ethics. (Please upload your review as an attachment if it exceeds

6. PLOS authors have the option to publish the peer review history of their article (what does this mean?). If published, this will include your full peer review and any attached files.

Reviewer #1: No

Reviewer #2: **Yes: **Amal Mahmoud Aboelmaaty

---

## [Author Response · Author response to Decision Letter 0]

10 Jan 2025

AU: Done

“Funding for this work was provided, in part, by the Virginia Agricultural Experiment Station and the Hatch Program of the National Institute of Food and Agriculture, U.S. Department of Agriculture.”

AU: Done

“The authors would like to express their appreciation to Ali Wood for her dedication to this work and contribution to the manuscript. Funding for this work was provided, in part, by the Virginia Agricultural Experiment Station and the Hatch Program of the National Institute of Food and Agriculture, U.S. Department of Agriculture.”

“Funding for this work was provided, in part, by the Virginia Agricultural Experiment Station and the Hatch Program of the National Institute of Food and Agriculture, U.S. Department of Agriculture.”

AU: Done

AU: Done

6. We notice that your supplementary table is included in the manuscript file. Please remove them and upload them with the file type 'Supporting Information'. Please ensure that each Supporting Information file has a legend listed in the manuscript after the references list.

AU: Done

Reviewers' comments:

Reviewer's Responses to Questions

Comments to the Author

1. Is the manuscript technically sound, and do the data support the conclusions?

Reviewer #1: Yes

Reviewer #2: Yes

AU: Thank you to the reviewers for all of their evaluations!

2. Has the statistical analysis been performed appropriately and rigorously?

Reviewer #1: Yes

Reviewer #2: Yes

AU: Thank you!

3. Have the authors made all data underlying the findings in their manuscript fully available?

Reviewer #1: No

Reviewer #2: Yes

AU: Data is now available and information is provided on the submission platform and in the cover letter.

4. Is the manuscript presented in an intelligible fashion and written in standard English?

Reviewer #1: Yes

Reviewer #2: Yes

AU: Thank you.

5. Review Comments to the Author

Reviewer #1: The manuscript is well written. The language is generally clear. However, I think some minor adjustments or corrections need to be made to improve the article.

AU: We wish to thank the reviewer for taking the time to evaluate our submission!

On line 36 to 38, that is the abstract section, you wrote "In the final two experiments".... Are they actually two separate experiments? I think you need to rewrite that sentence to improve clarity.

AU: The reviewer is correct – thank you for pointing out our mistake. The verbiage indicating two final experiments was left over from an old draft where the experiments were described differently. We have revised accordingly.

I think you need to improve the clarity of line 100 to 105. In line 104, you wrote "and adapted to use as base".... That sentence is not very clear.

AU: Thank you for pointing this out. We have revised to improve the clarity.

Line 224 to 225, That sentence is not necessary in the result section. You have already mentioned the aim in the introduction.

AU: Thank you for this suggestion. The sentence has been removed.

The first two paragraphs of the discussion section are absolutely not necessary. I think you should discuss your findings and briefly explain how they relate or differ from similar studies. You only gave an overview of the literature, and that is something to be done in the introduction section. The same applies to the first few sentences of the third paragraph. The correct discussion only started in line 307. I think you need to remove the first two and half paragraphs in the discussion and improve on the discussion.

AU: Thank you for this suggestion. We agree with the reviewer’s assessment and have revised this portion significantly. The quality of the discussion is greatly improved as a result of this suggestion.

Reviewer #2: No comment for the authors. The topic is novel and the languish is sound. The article can be accepted in the current form.

All questions above were answered . No additional comments for the author, including concerns about dual publication, research ethics, or publication ethics. (Please upload your review as an attachment if it exceeds

AU: Thank you very much for taking the time to evaluate our submission!

6. PLOS authors have the option to publish the peer review history of their article (what does this mean?). If published, this will include your full peer review and any attached files.

Do you want your identity to be public for this peer review? For information about this choice, including consent withdrawal, please see our Privacy Policy.

Reviewer #1: No

Reviewer #2: Yes: Amal Mahmoud Aboelmaaty

---

## [Editor Report · Decision Letter 1]

15 Jan 2025

Cumulus cell expansion, nuclear maturation and embryonic development of bovine cumulus-oocyte complexes matured in varying concentrations of follicular fluid

PONE-D-24-44555R1

Dear Dr. Michelle L Rhoads

We’re pleased to inform you that your manuscript has been judged scientifically suitable for publication and will be formally accepted for publication once it meets all outstanding technical requirements.

Kind regards,

Sanaz Alaeejahromi

Academic Editor

PLOS ONE
---

## [Editor Report · Acceptance letter]

30 Jan 2025

PONE-D-24-44555R1 

PLOS ONE

Dear Dr. Rhoads, 

I'm pleased to inform you that your manuscript has been deemed suitable for publication in PLOS ONE. Congratulations! Your manuscript is now being handed over to our production team.

Kind regards, 

on behalf of

Dr. Sanaz Alaeejahromi 

Academic Editor

PLOS ONE